# Explaining with Contrastive Phrasal Highlighting:
# A Case Study in Assisting Humans to Detect Translation Differences

**Eleftheria Briakou**[1]*, **Navita Goyal**[2], **Marine Carpuat**[2]
[1] Google, [2] University of Maryland
ebriakou@google.com, {navita, marine}@cs.umd.edu

## Abstract

Explainable NLP techniques primarily explain by answering *"Which tokens in the input are responsible for this prediction?"*. We argue that for NLP models that make predictions by comparing two input texts, it is more useful to explain by answering *"What differences between the two inputs explain this prediction?"*. We introduce a technique to generate *contrastive phrasal highlights* that explain the predictions of a semantic divergence model via phrase-alignment-guided erasure. We show that the resulting highlights match human rationales of cross-lingual semantic differences better than popular post-hoc saliency techniques and that they successfully help people detect fine-grained meaning differences in human translations and critical machine translation errors.

## 1 Introduction

A common strategy to explain the predictions of NLP models is to highlight salient tokens in their inputs (Li et al., 2016; Ribeiro et al., 2016; Lundberg and Lee, 2017). However, this is suboptimal for the many NLP tasks that require comparing and contrasting two or more pieces of text to predict a class or a similarity score, such as natural language inference (Bowman et al., 2015), semantic textual similarity (Agirre et al., 2012), or evaluation and quality estimation of text generation (Bojar et al., 2017; Ma et al., 2018, 2019; Mathur et al., 2020; Freitag et al., 2021b, 2022). At the same time, a long line of research in social sciences shows that human explanations are contrastive (Miller, 2019), i.e., humans do not explain why an event happened in a vacuum but instead compare what happened to a contrast case. For instance, to explain how a Greek sentence and a translation in English differ, it is more intuitive to show how they differ (as presented in Figure 1) than to highlight all salient Greek and English tokens without specifying how they relate to each other.

---
*Work done while at the University of Maryland.

Figure 1: The color-coded contrastive phrasal highlights explain meaning differences between a Greek sentence (EL) and its English translation (EN) from Wikipedia.

In this work, we introduce a post-hoc explainability technique to answer *"What differences between two inputs explain a prediction"?* for NLP regression models that produce a score encoding a relationship between two text inputs. We apply it to the task of explaining meaning divergences between a text in one language and its translation. We design a phrase-alignment-guided erasure strategy to explain the predictions of a semantic divergence model and highlight minimal phrase pairs in the two inputs that result in meaning differences.

We extensively evaluate our approach with both proxy and human-centered evaluations. Comparing against human explanations for meaning differences in English-French human translations, our approach matches human rationales better than popular post-hoc methods that highlight salient tokens independently (Ribeiro et al., 2016; Lundberg and Lee, 2017). Acknowledging the limitations of such proxy evaluations (Buçinca et al., 2020), we conduct two IRB-approved human studies to assess the usefulness of our contrastive phrasal highlights when crowd-sourcing challenging bilingual annotation tasks. We find that our approach helps annotators reliably discover fine-grained meaning differences in English-French and English-Spanish human translations and detect critical errors in Portuguese-to-English machine trans-

lation. We make our code and data publicly available: `https://github.com/Elbria/Ex-SemDiv`.

## 2 Background

We focus on *local* explanations which explain why a specific prediction was made, often to help humans rely on predictions appropriately in machine-in-the-loop use cases (Doshi-Velez and Kim, 2017).

**Highlight Explanations** Local explanations often consist of highlighting input features deemed important to a prediction. In glass-box settings, prior works seek to quantify feature saliency with model-internal signals such as gradients (Arras et al., 2016; Smilkov et al., 2017), contextual decompositions (Murdoch et al., 2018), or attention mechanisms (Choi et al., 2016). However, these highlights' ability to serve as explanations has been questioned (Jain and Wallace, 2019; Wang et al., 2020; Moradi et al., 2021), leading to work revisiting technical solutions and evaluations under which attention *could* be seen as an explanation (Wiegreffe and Pinter, 2019; Moradi et al., 2019; Tutek and Snajder, 2020). Another family of approaches relies on erasure strategies to explain the predictions of black box models. Li et al. (2016) quantify a token's importance as the difference in model confidence once it is erased from the input, with several following works exploring different token erasure schemes (Feng et al., 2018; Kim et al., 2020). LIME (Ribeiro et al., 2016) approximates the model's local decision boundary with linear models by erasing multiple tokens from a model's input, while SHAP (Lundberg and Lee, 2017) computes Shapley values by estimating the marginal contribution of each token across all possible erasures. We build on this line of work by designing an erasure strategy that identifies salient *contrastive phrasal pairs* rather than independent salient tokens.

**Contrastive Explanations** Motivated by social science research suggesting that humans explain with respect to an (implicit) contrast case (Miller, 2019), recent work has sought to *generate* the latter by producing counterfactual explanations: minimally edited versions of the input that change the model prediction. Performing such edits usually requires training dedicated editors (Ross et al., 2021), prompting language models (Paranjape et al., 2021), and accessing knowledge bases (Chen et al., 2021), among others (Li et al., 2020; Chemmengath et al., 2022). Closer to our work, an-

other family of approaches extracts *highlights* contrastively: Jacovi and Goldberg (2021) extend erasure-based approaches to identify which input features lead to the actual prediction vs. a contrastive prediction, while Yin and Neubig (2022) extend saliency methods to explain the generation of contrast cases for language generation tasks. Our work explains the predictions of models that compare two inputs, where identifying salient phrase pairs that differ across them provides a natural mechanism for contrastive explanations.

**Evaluation** Most current work in NLP adopts *proxy* evaluations that compare automatic explanations with human explanations of the gold label, with numerous datasets encoding explanations using highlights or free-form text (Wiegreffe and Marasovic, 2021). Others rely on *simplified* human tasks, such as simulatability, evaluating whether explanations help people predict a model's prediction more accurately (Hase and Bansal, 2020; Nguyen, 2018). Despite the attractiveness of the above evaluations, evidence is growing that they do not reliably indicate how useful explanations are in practice (Buçinca et al., 2020). Boyd-Graber et al. (2022) call for application-grounded evaluations to directly assess whether explanations help people complete an actual task, as suggested by work in HCI (Suresh et al., 2021; Liao et al., 2020). We heed that call by complementing a proxy evaluation with two application-grounded user studies.

**Explaining Translation Differences** Detecting translation differences is a core task in multilingual NLP, e.g., to predict machine translation errors (Rei et al., 2020), to study how humans translate (Zhai et al., 2020), or to understand how multilingual resources such as Wikipedia differ across languages (Miquel-Ribé and Laniado, 2018; Jiang et al., 2017). Automatic approaches often score the degree of (dis)similarity between an input and its translation to quantify machine translation quality (Zhang et al., 2019; Sellam et al., 2020; Rei et al., 2020) or to detect meaning divergences in human translations (Vyas et al., 2018; Briakou and Carpuat, 2020; Wein and Schneider, 2021). However, sentence-level scores can be hard to interpret and do not pinpoint how translations differ. This has been addressed by tagging text with human translation processes (Zhai et al., 2018), with dedicated, supervised models or with word-level quality estimation (QE) tasks (Specia et al., 2018,

2020, 2021), often addressed by applying post-hoc explanation methods to sentence-level predictors (Treviso et al., 2021; Rei et al., 2022). We augment this work by contributing unsupervised contrastive explanations of translation differences and testing whether they help annotate them more reliably.

## 3 Explaining Divergences with Contrastive Phrasal Highlights

We introduce a highlighting approach to explain the predictions of a model that compares and contrasts two text inputs. We hypothesize that phrase pairs provide more appropriate *cognitive chunks* (Doshi-Velez and Kim, 2017) than tokens as basic units of explanations. For instance, instead of highlighting tokens that differ in the English-French pair of Figure 2, it is more natural to explain that the English text refers to the Maple Leaf Forever as a *national anthem* as opposed to a *chant patriotique* (patriotic song) in French and that the English phrase *"for many years"* has no equivalent in French.

Building on token-level erasure-based explanation methods (Li et al., 2016), we design a phrase-alignment-guided erasure strategy to explain the predictions of a model (Briakou and Carpuat, 2020) that ranks bilingual sentence pairs based on the granularity of their semantic similarity (assuming $\mathbf{S}$ is an equivalent pair and $\hat{\mathbf{S}}$ a pair containing semantic divergences, the model $\mathcal{R}(\cdot)$ ranks them such that $\mathcal{R}(\mathbf{S}) > \mathcal{R}(\hat{\mathbf{S}})$). As shown in Figure 2, given two input texts, we first extract a set of candidate counterfactual inputs by masking a subset of phrase pairs (§3.1). Then, we explain the model's prediction by selecting the phrase pair whose erasure maximally increases the similarity score between the two inputs (§3.2).

### 3.1 Alignment-Guided Phrasal Erasure

We propose an erasure approach that takes into account the input's structure. Given a sentence pair $\mathbf{S}$, we start by extracting a set of candidate counterfactual instances by deleting a single phrase pair from $\mathbf{S}$. Given that erasing all possible phrase pairs for each sentence is computationally prohibitive, we restrict our search to deleting phrases that belong to an aligned phrase table, $\mathcal{P}$.

Our phrase pair candidates for erasure are derived from classical statistical machine translation techniques developed for extracting phrase translation dictionaries from parallel corpora (Och et al., 1999; Koehn et al., 2007). Given two texts, we de-

rive word alignments based on multilingual word embeddings (Jalili Sabet et al., 2020) and then extract all phrase pairs consistent with the word alignments. A phrase pair $(p_1, p_2)$ is consistent with the word alignment if $p_1$ and $p_2$ are each contiguous sequences of words in each language, where all alignment links that start within $p_1$ map to tokens within $p_2$ and vice versa. Unaligned words (i.e., words aligned to a `null` token) can be included in either the source ($p_1$) or the target ($p_2$) phrases. As a result, the extracted phrase pairs comprise not only equivalent but also include related but divergent phrase pairs (*"national anthem"* and *"chant patriotique"*) and unaligned phrase (*"served for many years"*), as seen in Figure 2.

### 3.2 Explaining by Minimal Contrast

Given the sentence pair $\mathbf{S}$ and its aligned phrase table $\mathcal{P}$, we first extract a set of contrast cases $\tilde{\mathcal{P}}$ consisting of all the phrasal pairs that, once erased, make the two inputs more equivalent, as measured by an increase in score larger than a margin $\epsilon$:

$$\tilde{\mathcal{P}} = \left\{ p \in \mathcal{P}, \text{ s.t. } \mathcal{R}\Big(\textbf{DEL}\big[\mathbf{S}; p\big]\Big) > \mathcal{R}\Big(\mathbf{S}\Big) + \epsilon \right\}$$

Since explanations should be minimal (Hilton, 2017; Lipton, 1990), covering only the most relevant causes to reduce cognitive load, we select as contrastive phrasal highlights the phrase pairs that correspond to minimal edits with a large impact on predictions, as follows:

$$\arg\max \left\{ \mathcal{R}\Big(\textbf{DEL}\big[\mathbf{S}; p\big]\Big) \cdot \mathcal{BR}\Big(\mathbf{S}, p\Big) \right\}$$

$$\mathcal{BR}(\mathbf{S}, p) = \begin{cases} e^{-\frac{|p|}{|\mathbf{S}|}}, & \text{if } \mathcal{R}\Big(\textbf{DEL}\big[\mathbf{S}; p\big]\Big) \geq 0 \\ e^{+\frac{|p|}{|\mathbf{S}|}}, & \text{otherwise} \end{cases}$$

where the first term $\mathcal{R}(\textbf{DEL}\big[\mathbf{S}; p\big])$ encourages the extraction of a contrastive phrasal highlight $p$ corresponding to a high score under the model (i.e., deleting this phrase pair yields a contrast case), while the second term, $\mathcal{BR}(\mathbf{S}, p)$ corresponds to a *brevity reward* that encourages extraction of shorter highlights. $|\mathbf{S}|$ is computed by adding the length of each of the two sentences, and $\textbf{DEL}[\cdot]$ is a function that erases a phrase pair $p$ from $\mathbf{S}$.

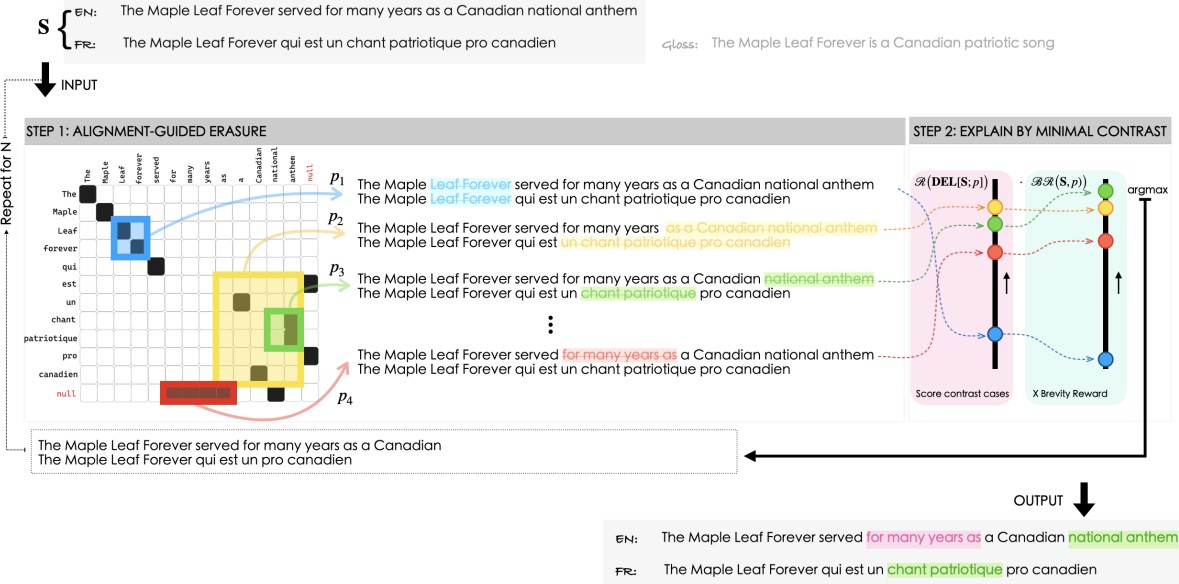

Figure 2: Our approach takes as input a sentence pair ($\mathbf{S}$) and extracts a set of perturbed inputs by erasing phrasal pairs guided by word alignments (Step 1). Then it explains the prediction of a regressor $\mathcal{R}(\mathbf{S})$ by highlighting the phrasal pair $p$ that, once deleted, maximizes the model's prediction $\mathcal{R}(\text{DEL}[\mathbf{S}; p])$ multiplied by a brevity reward $\mathcal{BR}(\mathbf{S}, p)$ that encourages the extraction of short phrasal pairs (Step 2).

The above approach identifies a single phrase pair that explains the divergence prediction for the original sentence $\mathbf{S}$. To extract multiple explanations, we repeat this process by iteratively erasing the extracted contrastive phrasal highlight from the current input sentence $\mathbf{S}' = \text{DEL}\big[\mathbf{S}; p\big]$, and repeat the steps in §3.1 and §3.2. This iterative process ends when, for a given input, none of the extracted counterfactual instances yield a more equivalent pair, i.e., $\tilde{\mathcal{P}} = \varnothing$, or we reach an equivalent input under the divergent ranker, i.e., $\mathcal{R}(\mathbf{S}') > 0$.

## 4 Proxy Evaluation

In this section, we describe our proxy evaluation based on human-provided highlights. We acknowledge that proxy evaluations encounter validity issues (Boyd-Graber et al., 2022) and use them primarily to guide system development and validate against standard highlighting methods.

### 4.1 Experimental Setup

**Explanandum**   We seek to explain the prediction of a divergence ranking model that is trained as recommended in prior work (see Appendix A.1).

**Explainers**   We contrast our contrastive phrasal highlights against a RANDOM baseline that highlights tokens in each sentence at random with $0.5$ probability, and two standard post-hoc explanation

methods, LIME and SHAP, that seek to explain the predictions of the explanandum.

**Reference Highlights**   We evaluate our approach using the REFRESD dataset,[1] which is manually annotated with divergences of fine-grained and coarse-grained granularity, along with rationales that justify the sentence label. We compare our contrastive phrasal highlights with the human rationales on a subset of $418$ challenging instances annotated as having "Some Meaning Differences" at the sentence level, where we expect the more subtle differences in meaning to be found.

**Evaluation Metrics**   Following prior proxy evaluations of explanations (Ross et al., 2021; DeYoung et al., 2020), we compute: 1. *Agreement with Human Rationales*: Precision, Recall, and F-1 scores computed against human rationales; 2. *Minimality*: the length of the contrastive phrasal highlights, measured as the number of highlighted tokens.

### 4.2 Results

As seen in Table 1, explaining divergence predictions by extracting contrastive phrasal highlights significantly outperforms both LIME and SHAP. Those standard highlighting baselines even under-

---

[1]https://github.com/Elbria/xling-SemDiv/tree/master/REFreSD

|  | ENGLISH | | | | FRENCH | | | |
|---|---|---|---|---|---|---|---|---|
|  | PR. | RE. | F1 | DEL. | PR. | RE. | F1 | DEL. |
| ORACLE | 100 | 100 | 100 | 39% | 100 | 100 | 100 | 43% |
| RANDOM | 39 | 48 | 43 | 50% | 42 | 49 | 45 | 50% |
| LIME | 45 | 37 | 41 | 30% | 44 | 37 | 40 | 34% |
| SHAP | 53 | 34 | 41 | 25% | 50 | 32 | 39 | 26% |
| OURS (− BR) | 52 | 76 | 62 | 54% | 56 | 74 | 64 | 55% |
| OURS | 58 | 61 | 62 | 37% | 60 | 55 | 64 | 37% |

Table 1: Proxy evaluation results with respect to human rationales on REFRESD.

Table 2: Examples of divergence explanations (**HUMAN** corresponds to REFRESD rationales).

perform the RANDOM baseline.[2] A closer look at the outputs (Table 2) indicates that both LIME and SHAP suffer from two major issues: sparsity and accuracy. We attribute those issues to the fact that both approaches operate in token space, ignoring the interdependent relationships between tokens across the two languages. By explicitly modeling such relationships, our approach produces highlights that match the ones in REFRESD better.

We measure the impact of the brevity reward by ablation (i.e., OURS (− BR)). This achieves the highest recall at the expense of precision: dropping the brevity reward produces fewer and longer highlights per instance. It highlights more than 50% of each sentence pair on average, while humans only highlight about 40%. The brevity reward thus helps match reference highlights better, supporting the benefits of producing minimal explanations.

In sum, these results are promising indicators that contrastive phrasal highlights could be helpful when detecting semantic divergences. However, as highlighted by recent work on explanation evalu-

ation, proxy evaluations can be misleading. Given that prior work evaluates highlight-based explanations of (machine) translation errors based *solely on automatic evaluations*, we chose to initiate a human-centered discussion that evaluates the usefulness of such highlights directly with an in-depth exploration of multiple small-scale *application-grounded* evaluations as detailed below.

## 5 Application-Grounded Evaluation I: Annotation of Semantic Divergences

As shown by prior work, annotating fine-grained divergences is a challenging task and requires dedicated annotation protocols based on human rationales (Briakou and Carpuat, 2020) or abstract meaning representation frameworks (Wein et al., 2022) to achieve moderate agreement. As a result, such annotations are usually hard to collect with crowd workers since they require explicit annotator training. In this section, we ask whether contrastive phrasal highlights help crowd-workers annotate divergences more reliably, which could ease the need for complex and expensive annotation protocols. Concretely, we test the following hypothesis: *Contrastive phrasal highlights improve the annotation of fine-grained semantic divergences in terms of accuracy and agreement.*

---

[2]Since LIME and SHAP are feature attribution methods that assign a continuous score to each word, we checked that changing the threshold from the default $t = 0$ does not improve the results, as both methods suffer from a precision-recall tradeoff.

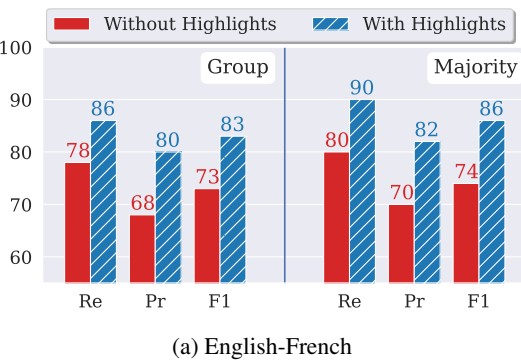

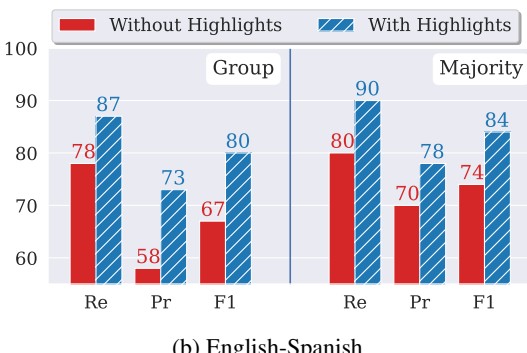

(a) English-French  (b) English-Spanish

Figure 3: Annotation-grounded evaluation comparing with- vs. without-highlights annotation conditions. Contrastive phrasal highlights significantly ($p = 0.1$) improve the annotation of fine-grained semantic divergences.

## 5.1 Experimental Setup

**Explanandum**   We seek to explain the predictions of Divergent mBERT. We train separate models for English-French and English-Spanish following the approach detailed in Appendix A.1.

**Study Data**   To provide controlled yet realistic test samples, we construct a dataset of fine-grained divergences that mimic translation processes used by human translators (Zhai et al., 2019). We start with English-French and English-Spanish human translations from the multi-parallel FLORES benchmark (Goyal et al., 2022). We randomly select 50 *seed* translations among those identified as semantically equivalent pairs by Divergent mBERT. Then, we introduce fine-grained meaning differences by editing the English references of 10 of those samples. We perform edits motivated by translation processes, such as modulation, explicitation, and reduction (Zhai et al., 2019), by interaction with ChatGPT.[3]  For instance, we introduce a generalization as follows:

> Despite leaving the show in 1993 he kept *[the title of executive producer] → [a senior role]* and continued to receive tens of millions of dollars every season in royalties.

To create difficult examples of equivalent translations, we also paraphrase the English references of 10 seed examples with ChatGPT to introduce syntactic changes. We validated all examples, ensuring that edits actually match the intended divergences in the first case (Figures 6 and 7) and that syntactic changes do not alter meaning in the second (Figure 8). The final dataset consists of 35 semantically equivalent translations (split into 25 original trans-

lations and 10 syntactically divergent paraphrases) and 15 fine-grained divergences resembling translation processes.

## 5.2 Study Design

We ran a controlled evaluation across the two language pairs (English-French and English-Spanish). We describe the task and study design below.

**Task Description**   An annotation instance is a sentence pair in English and, e.g., French. Bilingual speakers are asked to read each sentence pair closely, determine whether the *"sentence pair contains any difference in meaning"*, and finally characterize the difference as "Added" vs. "Changed"."Added" refers to added information, while "Changed" refers to information that is present in both languages but does not precisely match—mirroring prior annotation divergence protocols (Briakou and Carpuat, 2020). We include a screenshot of the task description presented to annotators in Figures 5 and 7 of Appendix A.4.

**Conditions**   We study two conditions: one in which participants are shown highlights ( ✓ HIGHLIGHTS ) and a second in which they are not ( ✗ HIGHLIGHTS ). The only information available to participants about the highlights is that they *"are AI-generated and indicative of meaning differences"*. We emphasize that highlights are not presented as explanations of divergence predictions, as we want annotators to choose how and whether they use them in their assessments based on their intuitions.

**Procedures**   We conduct a between-subjects study where participants are randomly assigned to a condition. Participants are first presented with a tutorial that explains the task and relevant ter-

---

[3]Details of the ChatGPT prompts are in Appendix A.3.

minology. Each participant is presented with 25 instances from one of the two studied conditions. Each batch of annotated instances contained $30\%$ of semantically divergent edits, $20\%$ of syntactically divergent edits, and $50\%$ original semantically equivalent pairs. Instances within each batch are randomized. We include two attention checks where participants are asked to indicate their answers to the previous question. After completing the task, participants were asked to complete a brief survey assessing their perception of the generated highlights (if assigned to this condition) and finally were asked to provide their free-form feedback on the study along with demographic information, such as gender and age. The average time of the study was 20 minutes. In sum, we collect 3 annotations per instance and annotate a total of 100 instances (50 per condition) for each of the two language pairs studied.

**Participants**   We recruit 12 participants per language pair using Prolific.[4] Each participant is restricted to taking the study only once. None of them failed both attention checks; hence we did not exclude any of the collected annotations from our analysis. All participants identified as bilingual speakers in the languages involved in each study. Participants are compensated at 15 USD per hour.

### 5.3   Measures

Our main evaluation measures are Precision, Recall, and F1 computed by assuming that semantically edited instances correspond to divergences, while the rest are treated as semantically equivalent pairs. We report those accuracy statistics both at the group level and also by majority voting annotations. Furthermore, we summarize the responses provided as free-form feedback and report subjective measures that reflect participants' perceived understanding of the explanations, i.e., *"the highlights are useful in detecting meaning differences"*, if provided. The latter measures are collected on a 5-point Likert scale. We report statistical significance based on bootstrap resampling. We draw samples randomly from the collected annotations with replacement. The sample size is the same as the one of the collected annotations, while the number of resamples is set to 1,000. For the application-grounded evaluation I, which contains a total of 150 annotations per language, the significance level is $p = 0.1$.

---

[4]https://www.prolific.co/

### 5.4   Study Results

**Reliability of Annotations**   As shown in Figure 3, the bilingual *group* that annotated fine-grained meaning differences in the presence of contrastive explanations—✓ HIGHLIGHTS , is significantly ($p = 0.1$) more accurate across Precision, Recall, and F1 scores and both language pairs, compared to ✗ HIGHLIGHTS . Furthermore, those improvements carry over when aggregating annotation results by *majority* voting annotations across instances. As a result, this leads us to accept the hypothesis that contrastive phrasal highlights improve the annotation of fine-grained divergences by bilingual speakers. Finally, detecting divergences with highlights additionally improves the reliability of annotations are measured by Cohen's Kappa inter-annotator agreement statistics:

|  | EN-FR | EN-ES |
|---|---|---|
| ✗ HIGHLIGHTS | 0.51 (moderate) | 0.33 (fair) |
| ✓ HIGHLIGHTS | 0.66 (substantial) | 0.52 (moderate) |

**Subjective Measures & User Feedback**   Overall, bilingual speakers presented with contrastive phrasal highlights agreed they were useful in helping them spot fine-grained divergences—average self-reported usefulness of 3.8 for EN-FR and 4.2 for EN-ES. Finally, although contrastive phrasal highlights were useful, annotators also note that they cannot entirely rely on them. For instance, some of the participant's feedback is *"The highlights are useful but not $100\%$ reliable, that is because I found other added/changed words that AI did not highlight"* and *"In most cases, the words highlighted by the AI have helped to detect possible differences"*.

## 6   Application-Grounded Evaluation II: Critical Error Detection

In this section, we examine the potential of using contrastive phrasal highlights to assist bilingual humans in detecting critical errors in machine translation outputs. Recent work on quality estimation for machine translation proposes an error-based evaluation framework where bilingual *professional* annotators are asked to: highlight translation errors; rate their severity as major or minor; and finally, indicate their type (e.g., mistranslation vs. terminology, etc.) (Freitag et al., 2021a). Drawing on those intuitions, we aim to study whether a simplified evaluation framework that uses contrastive

phrasal highlights yields reliable annotations of critical errors with bilingual *crowd workers*. This is a hard task as these errors are rare in high-quality systems and might be missed when parsing translations quickly. In what follows, we test the hypothesis: *Contrastive phrasal highlights help bilingual crowd-workers detect critical (accuracy) errors in machine translation outputs.*

## 6.1 Experimental Setup

**Explanandum**   We explain the predictions of Divergent mBERT trained for English-Portuguese following the process detail in Appendix A.1. We emphasize that the explanandum is sensitive to detecting any meaning differences and not only critical errors, and therefore, we expect the contrastive phrasal highlights to surface both minor and major accuracy errors.

**Study Data**   For the purposes of the study, we use the synthetic Portuguese-English dataset from the Critical Error Detection task of WMT (Freitag et al., 2021b). The dataset consists of pairs sampled from the OPUS parallel corpus (Tiedemann, 2012) treated as "not containing a critical error". Then, the WMT organizers artificially corrupted $10\%$ of those pairs to introduce critical errors that reflect hallucinated content, untranslated content, or mistranslated texts. We will refer to those errors as WMT for the rest of the paper. Additionally, to make the task hard for humans, we a) filter out WMT critical errors that concern deviation in numbers, time, units, or dates and b) artificially introduce *negation* errors: we edit the English text locally by changing one or two words in such a way that the meaning of the sentence is entirely flipped (e.g., the word *benefits* in an English reference is replaced by the word *harms*). In total, the study dataset consists of 30 translations that do not contain a critical error (also detected as equivalent under the explanandum), 10 translations reflecting WMT errors, and 10 translations reflecting local *negation* errors.

## 6.2 Study Design

We test our hypothesis with a user study of critical error detection in English translations of Portuguese texts, as described below.

**Task Description**   An annotation instance is an excerpt in Portuguese and its (machine) translation in English. Bilingual speakers are asked to read the two excerpts, determine whether the translation contains an accuracy error, and then rate its severity

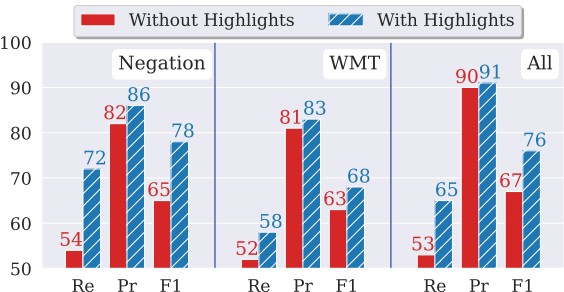

Figure 4: Application-grounded evaluation results comparing ✗ HIGHLIGHTS vs. ✓ HIGHLIGHTS annotation conditions. Contrastive phrasal highlights significantly improve the Recall and F1 scores when detecting *negation* errors ($p = 0.05$) and errors in the ALL set. Precision improvements across sets and any improvements on the WMT set are not significant ($p > 0.05$).

as being *minor* or *major*. Following Freitag et al. (2021a), minor errors are defined as errors that do not lead to loss of meaning and would not confuse or mislead the reader, while major ones may confuse or mislead the reader due to significant changes in meaning. We ask participants to factor out fluency issues in their assessment as much as possible. We include a screenshot of the task description presented to annotators in Figures 9 and 10 of Appendix A.4.

**Conditions & Evaluation Constructs**   We use Precision, Recall, and F1 as our main evaluation measures and follow the same conditions and subjective measures described in §5. The above scores are calculated only against major errors, as those are the ones for which we have gold labels. We report statistical significance based on bootstrap resampling as outlined in §5.3. For this study, which contains a total of 250 annotations, the significance level is $p = 0.05$.

**Procedures & Participants**   We collect 5 assessments per instance and annotate a total of 100 instances (50 per condition). Each participant is presented with 25 instances, among which $20\%$ correspond to WMT critical errors, $20\%$ to negation errors, and the rest to original OPUS parallel texts. We recruit a total of 40 participants, all of whom self-identified as proficient in English and Portuguese. All other participant recruitment details and study procedures are the same as in §5.

## 6.3 Study Results

**Main Findings**   As shown in Figure 4, the user group presented with contrastive phrasal highlights

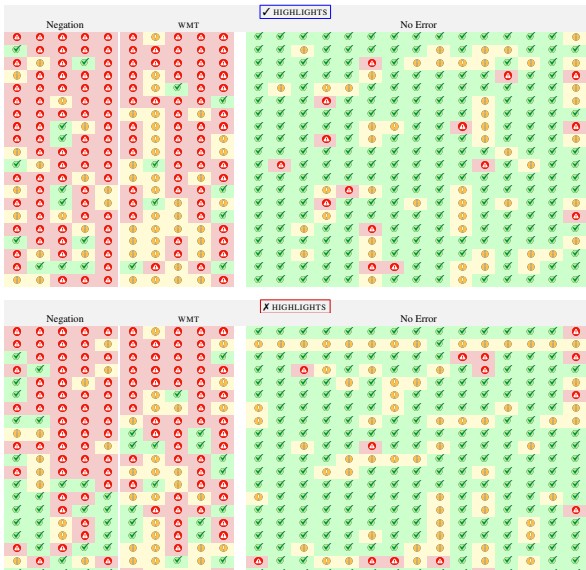

Table 3: Critical Error Detection user study results. Each row represents an annotator, and each cell is an annotation, where ✔ indicates no accuracy error detected, ⚠ indicates a minor error, and ⛔ a critical error.

exhibits a higher recall in detecting critical errors compared to the one that does not access them. Crucially, we report significant improvements ($p = 0.05$) in detecting *negation* errors, which are harder to spot. A closer look at the differences between the two error classes reveals that detecting negation errors is more challenging as annotators have to pay close attention to the compared texts and the relationships between them to spot local mistranslations that cause a major shift in meaning. On the other hand, WMT errors are more easily spotted as they mostly represent detached hallucination phenomena (i.e., an entire phrase is added to the English translation), which we hypothesize can be implied by length differences of the compared texts. Our full annotation results, presented in Table 3, further validate our hypothesis: WMT errors are not genuinely missed out but rather rated as minor by some annotators. On the other hand, usually, negation errors are not only misperceived as minor but are entirely overlooked. We include examples of annotations in Figure 10.

**Subjective Measures & User Feedback** Overall, bilingual speakers found the highlights helpful, with an average self-reported usefulness of 3.9. Additionally, they reported they would like to use them as a tool to assist them in detecting critical errors, with an average score of 3.8. We include details on the distribution of ratings in Table 4. Last, a

closer look at the users' feedback sheds some light on our approach's current strengths and limitations. Although highlights were in principle useful and a *"good feature to have especially when proofreading a great amount of translated texts"*, some annotators spotted *"a high percentage of false positives"* (see Figures 10i and 10j in Appendix A.4) and raised concerns that *"relying solely on them might make them less aware of errors hidden in non-highlighted texts"* (see Figure 10b in Appendix A.4).

## 7 Conclusion

We introduce an approach to extracting contrastive phrasal highlights that explain NLP models that take as input *two* texts. Unlike existing techniques that highlight input *tokens* independently, we consider the input's structure and explicitly model the relationships between contrasting inputs.

We study the effectiveness of contrastive phrasal highlights by explaining the predictions of a divergence ranker that compares and contrasts meaning differences in bilingual texts. Our proxy evaluation confirms our approach outperforms standard explainability approaches, that highlight tokens, by matching human-provided rationales of divergences in English-French Wikipedia texts. Finally, through a series of human-centered evaluations, we explore the usefulness of contrastive phrasal highlights in application-grounded contexts for three language pairs. Our results suggest that contrastive phrasal highlights assist bilingual speakers in detecting fine-grained meaning differences in (human) translated texts and critical errors due to local mistranslations in machine-translated texts.

Our findings create opportunities for designing better machine-in-the-loop pipelines to identify critical machine translation errors grounded in high-stake settings, study translation data at scale, and facilitate the creation of multilingual content through crowd-based efforts.

## Acknowledgements

We thank Sweta Agrawal, Hal Daumé III, Luke Zettlemoyer, Philip Resnik, Leo Zhicheng Liu, the anonymous reviewers, and the CLIP lab at UMD for helpful comments. This work was funded in part by the U.S. Army Grant No. W911NF2120076 and by the National Science Foundation under Awards No. 1750695 and 2147292. Any opinions, findings, and conclusions or recommendations expressed in this material are those of the authors.

## Limitations

Our work contributed evidence that contrastive phrasal highlights can provide a framework that assists humans in detecting meaning differences of a specific nature (i.e., fine-grained translation processes or local critical errors). However, detecting translation differences in the wild requires covering the entire distribution of meaning differences (Freitag et al., 2021a), which we leave for future work.

Moreover, although our method builds on unsupervised modules (i.e., Divergent mBERT and SimAlign) that do not rely on supervised data and can, in principle, be applied to any language pair, we have only evaluated our approach in high-resource settings where we expect both the explanandum and the word alignment models to be reasonably accurate. Therefore, further work should be conducted to explore how our findings generalize to other settings, such as in low-resource regimes where we might expect the alignment to be of poorer quality, introducing errors that may impact humans' perceived understanding of the contrastive highlights differently.

Additionally, future work should explore how the alignment module should be operationalized for other NLP tasks that take as input two texts, potentially exploring monolingual aligners (Lan et al., 2021) or additional structured information such as abstract meaning representations (Banarescu et al., 2013).

Finally, the significance of our results could be strengthened by increasing the sample sizes of future user studies. We view our current findings as solid starting points for an in-depth exploration of the usefulness of highlights for human-centered applications that future studies can build upon and extend with broad explorations.

## Ethics Statement

Both studies are approved by the University of Maryland Institutional Review Board (IRB number $2018458 - 1$). As discussed in the paper, all participants involved in our studies gave their consent and were compensated at a rate of 15 USD per hour. We collected minimal demographic information, and participants could opt out of answering the demographic questions. We did not collect any further potentially identifiable information. Participation in our study was completely voluntary, and participants had the right to withdraw at any time.

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

## A  Appendices

### A.1  Explanandum Training Details

The divergence ranking model is trained using the public implementation of Divergent mBERT (Briakou and Carpuat, 2020). Synthetic divergences are generated starting from the 5,000 top-scoring WikiMatrix sentences based on LASER score (i.e., seed equivalents). We fine-tune the "BERT-Base Multilingual Cased" model (Devlin et al., 2019) and set the margin equal to 5 as per the original implementation.

### A.2  Subjective Measures

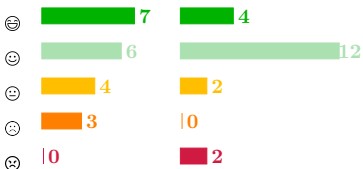

Table 4: Bilingual speakers' agreement ratings with the statement "Highlights were useful in helping me detect meaning differences in the compared texts" (left) and "I would like to use the highlights to help me detect critical errors" (right).

### A.3  ChatGPT Details

Starting from a given English sentence, we asked ChatGPT to edit the sentence in a way that introduces a small meaning difference. Below, we include the different prompts we used:

- *"Can you edit a small phrase in sentence X to introduce a small meaning difference?"*

- *"Can you delete/add a phrase in sentence X to introduce a small difference in meaning?"*

- *"Can you edit a small phrase that makes sentence X more general/explicit?"*

We then reviewed the edits introduced by ChatGPTto make sure it indeed introduced small meaning differences. If we are not satisfied by a current edit, we ask again and potentially specify what parts of the sentence we would like to be edited.

### A.4  User Study Interfaces

Below we include screenshots of our user study interfaces along with annotation examples for each of our evaluation tasks.

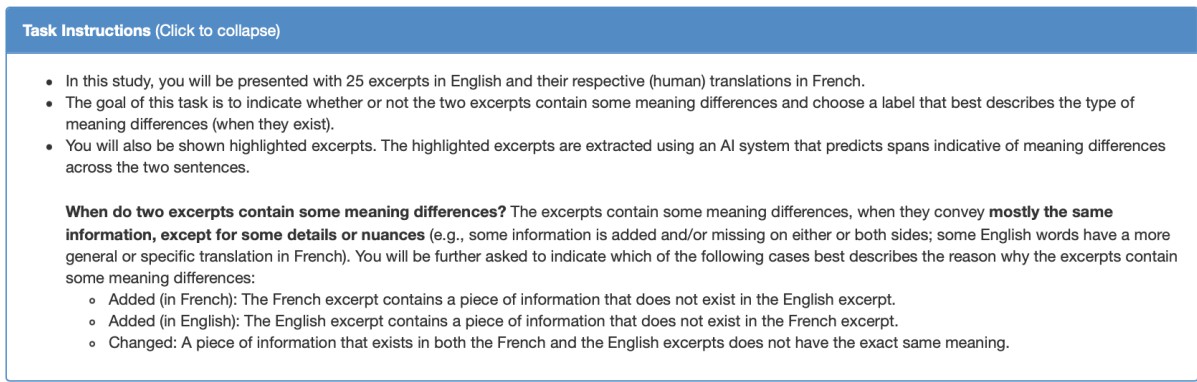

Figure 5: Instructions for application-grounded evaluation I: Annotation of Semantic Divergences.

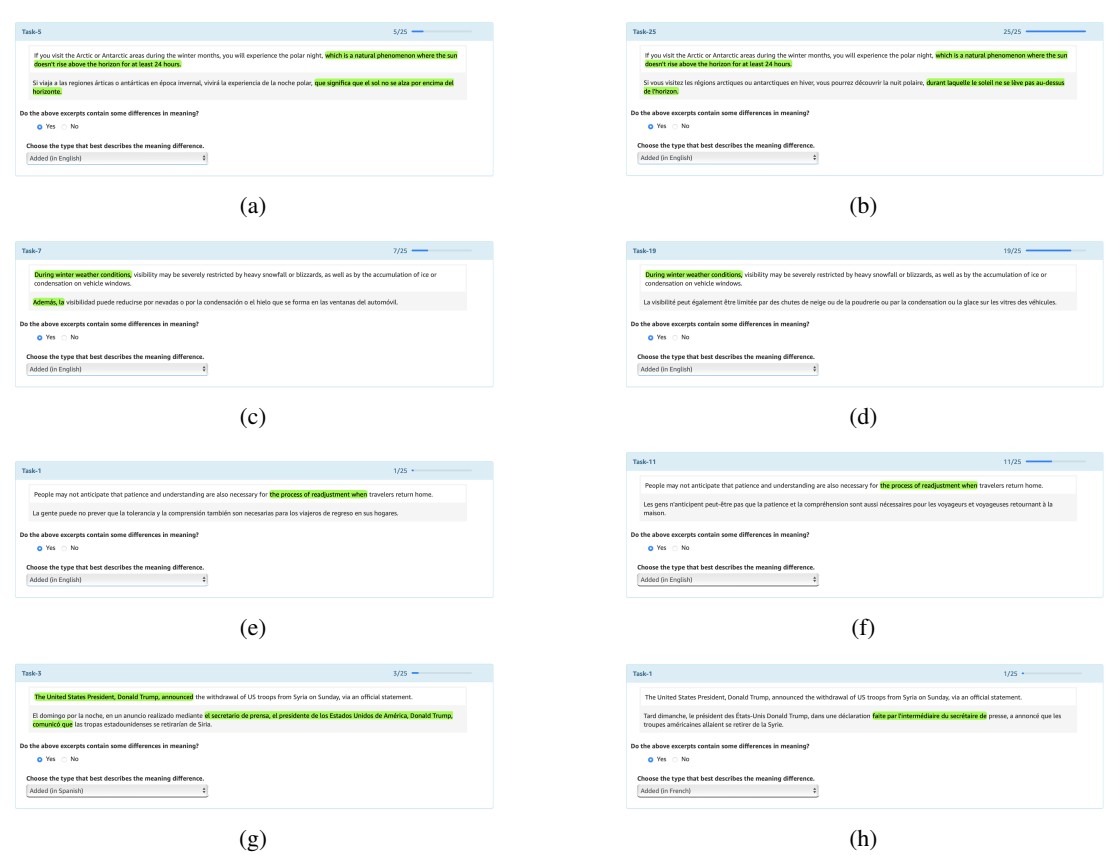

Figure 6: Annotations of fine-grained semantic divergences (English-Spanish on the left and English-French on the righ) reflective of *explicitation* and _**reduction**_ translation processes. Contrastive highlights subsume added content that is present in one language but missing from the other.

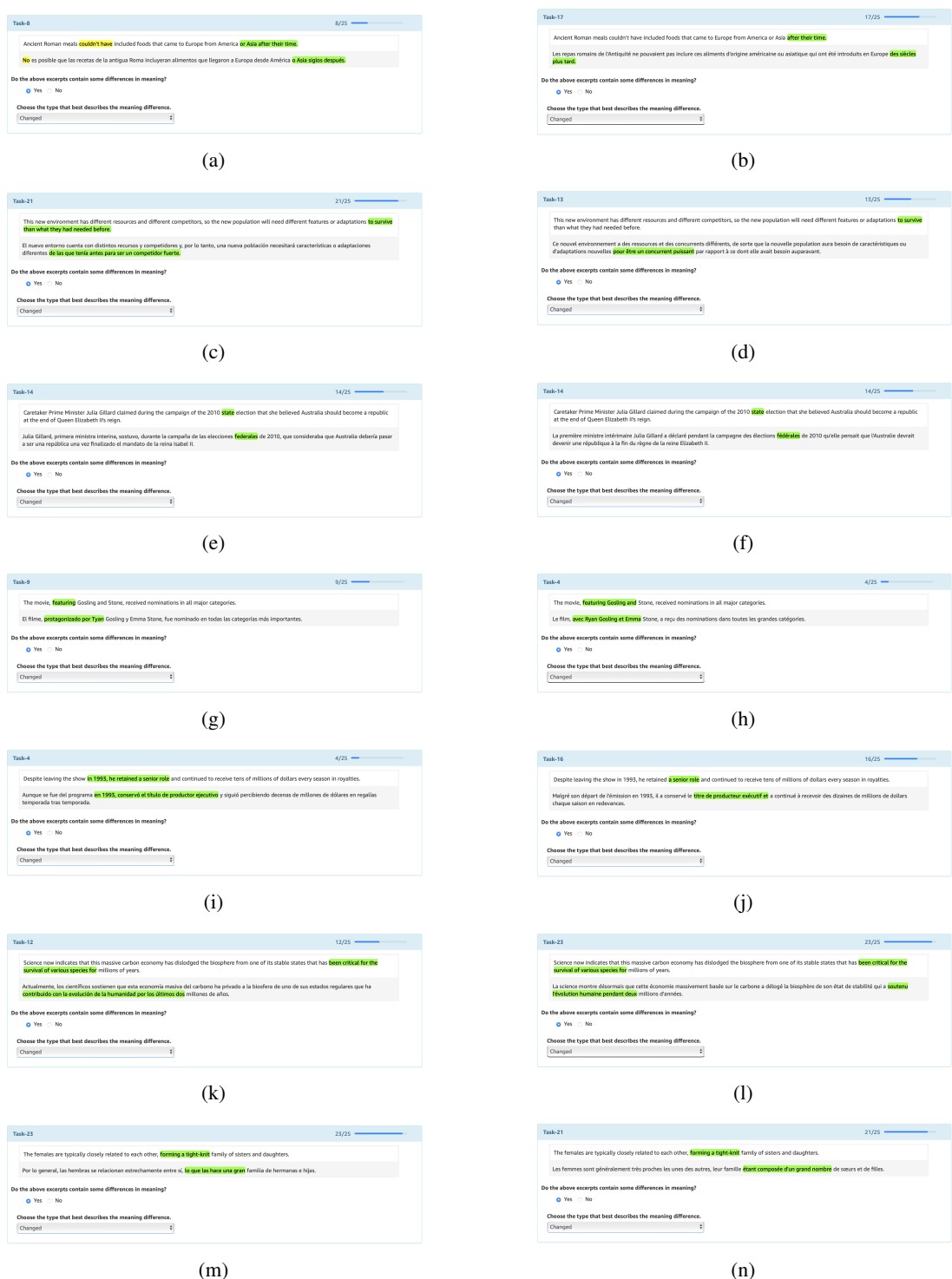

Figure 7: Annotations of fine-grained semantic divergences (English-Spanish on the left and English-French on the right) reflective of ***modulation*** translation processes. Contrastive highlights frequently subsume content that does not convey the exact same meaning across languages.

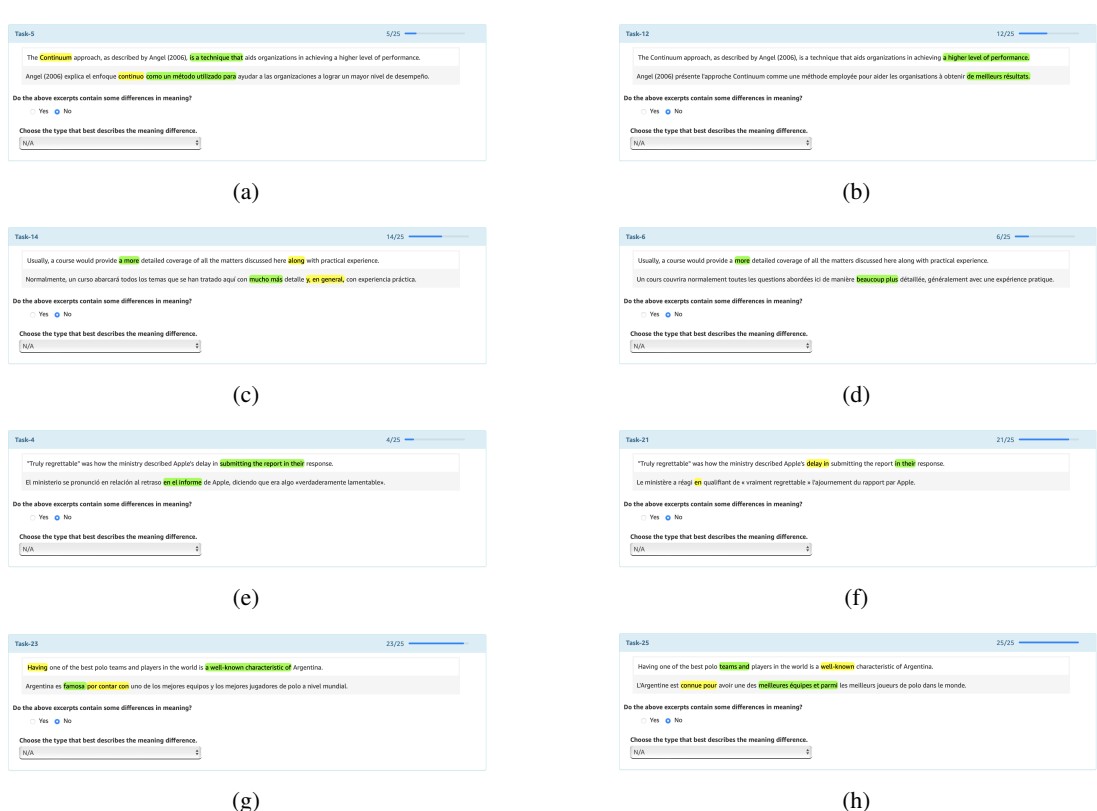

Figure 8: Annotations of semantically equivalent pairs (English-Spanish on the left and English-French on the right) that reflect syntactic divergences. Contrastive highlights surface *false positive* segments.

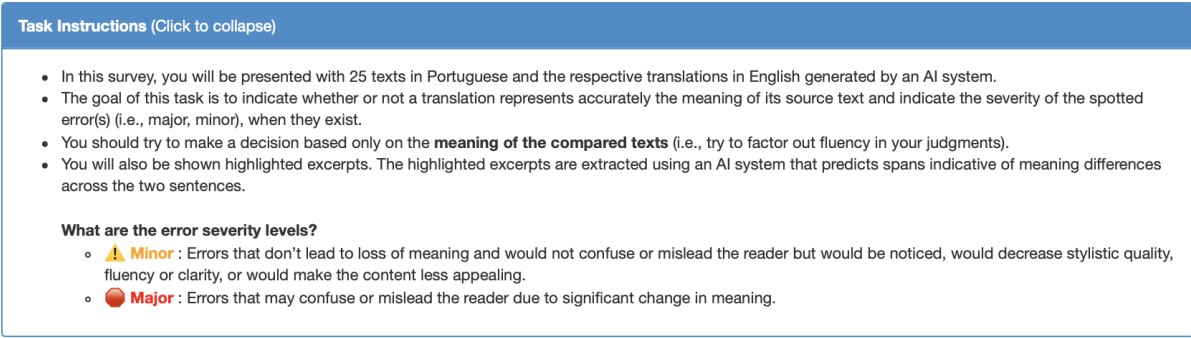

Figure 9: Instructions for application-grounded evaluation II: Critical Error Detection.

(a) **Major error** (negation)

(b) **Major error** (negation)

(c) **Major error** (negation)

(d) **Major error** (hallucination)

(e) **Major error** (hallucination)

(f) **Major error** (hallucination)

(g) **Minor error**

(h) **Minor error**

(i) **No error**

(j) **No error**

Figure 10: Annotations of accuracy errors in Portuguese translations of English texts. Contrastive highlights frequently surface meaning differences across the compared texts.