# OpenReview forum: "Explaining with Contrastive Phrasal Highlighting: A Case Study in Assisting Humans to Detect Translation Differences"
_EMNLP/2023/Conference — EMNLP 2023 Main_

### Official Review · Reviewer_WUTA · 2023-08-07

**Soundness:** 3

**Excitement:**

4: Strong: This paper deepens the understanding of some phenomenon or lowers the barriers to an existing research direction.

**Missing References:**

[1] Arras et al., 2017, “What is relevant in a text document?”: An interpretable machine learning approach

[2] Moradi et al., 2021, Measuring and Improving Faithfulness of Attention in Neural Machine Translation, EACL 21

[3] Murdoch et al., 2018, Beyond word importance: Contextual decomposition to extract interactions from LSTMs, ICLR 2018

[4] Moradi et al., 2019, Interrogating the explanatory power of attention in neural machine translation

**Paper Topic And Main Contributions:**

Taking a departure from the conventional token-relevance based approaches for model explanability, this paper proposes to focus on the differences between two input sentences to explain an output translation. The paper introduces phrase alignment guided erasure to generate contrastive phrasal highlights to explain predictions of semantic divergence model. This augments post-hoc explanation methods by contributing unsupervised contrastive explanations of translation differences.

The basic design follows a phrase-alignment guided occlusion strategy that ranks bilingual sentence pairs based on the degree of their semantic similarity. Given two input text, a set of candidate counterfactual inputs is extracted by masking a subset of phrase pairs using a statistical MT based phrase table. Then, the model's prediction is explained by selecting the phrase pair whose omission maximally increases the similarity between the two inputs. To guide this, two measures are proposed - 1. a penalty term that encourages the extraction of a high contrast phrase pair under the model, and 2. a brevity reward that encourages shorter phrases.

Finally, instead of relying fully on proxy evaluation, application-grounded evaluation measures are proposed.

**Questions For The Authors:**

1. Are the highlights always continuous spans? How do handle the scenarios where the alignments are not at the phrase level and are non-continuous spans?

**Reasons To Accept:**

Unlike existing techniques for MT explainability that highlight tokens independently, the proposed contrastive phrasal-pair erasure takes into account the input's structure and explicitly model the relationship b/w contrasting inputs. The results suggest that contrastive phrasal highlights assist bilingual speakers in detecting fine-grained differences in translated texts and other critical mistakes due to mistranslations. This creates opportunities for designing better machine-in-the-loop pipelines to detect MT errors.

The paper is well written and easy to follow. The experiments are thorough with ample analysis.

**Reasons To Reject:**

There are no major reasons to reject. However, based on the prerequisite of constructing a phrase table for the technique to work, the use of this approach in low-resourced scenarios might be limited.

The approach relies on divergent sentences for fine-grained meaning differences. This likely makes the approach sensitive to the quality of this candidate as paraphrase models are imperfect, especially when the input is domain specific.

**Reproducibility:**

3: Could reproduce the results with some difficulty. The settings of parameters are underspecified or subjectively determined; the training/evaluation data are not widely available.

**Reviewer Confidence:**

4: Quite sure. I tried to check the important points carefully. It's unlikely, though conceivable, that I missed something that should affect my ratings.

---

> ### Author Rebuttal · Authors · 2023-08-28
>
> We thank the reviewer for their feedback and for providing the missing references. In the revised version of our paper, we will incorporate a discussion of those works into our Background section.
>
> &#8594; **Applicability to low-resource languages:** We agree with the reviewer that the application of our work, as well as the application of any quality estimation model, is limited in low-resource scenarios. This is something we additionally acknowledge and discuss in our Limitations section. However, despite those limitations, we also argue there is a greater necessity for the implementation of our approach in high-resource settings where machine translation is reasonably fluent and local critical errors are more likely to be missed by humans. On the other hand, low-resource machine translation typically suffers more from fluency issues that, to some extent, are also indicative of other adequacy issues. Thus, it remains unclear whether highlights-based explanations alone are useful signals in low-resource settings to start with or whether other types of explanations are needed.
>
> &#8594; **Reliance on explainer:** We agree with the reviewer that by problem construction, the success of our approach, i.e., the quality of the produced highlight-based explanations, is conditioned on the success of the underlying explainer. However, this is an issue with *any* explainability technique, including the prior work we compare our approach with. Additionally, although this is not an application we explored in the current paper, our explanations could potentially be used to expose issues with the explainer and further improve it.
>
> &#8594; **Q1. Handling non-contiguous spans:** This is correct; our highlights are always continuous spans as we use Simalign as our aligner, which does not handle alignments between non-continuous spans. However, in principle, such cases could be addressed within our pipeline if we replace Simalign with a non-contiguous alignment-based model.

---

### Official Review · Reviewer_w8E1 · 2023-08-09

**Soundness:** 3

**Excitement:**

4: Strong: This paper deepens the understanding of some phenomenon or lowers the barriers to an existing research direction.

**Paper Topic And Main Contributions:**

In this paper, the authors consider the problem of identifying differences in translated sentences. These differences may be produced by errors in statistical MT systems, by crowdsourced (or other non-professional) translators, etc. In particular, the work in this paper focuses on identifying the parts of the sentences in a translation pair that correspond to the semantic differences, while leaving semantically-equivalent parts alone.

The authors' method for this relies on pre-existing phrasal alignment packages to produce what they call *contrastive phrasal highlights*, where the focus is on finding relevant phrases rather than individual tokens. The method is relatively straightforward, which I see as a strength. The authors compare their method against previous methods by evaluating against human judgements, finding that contrastive phrasal highlighting produces results that match human judgements much more closely.

The authors then follow this up with two application-grounded studies involving bilingual crowdworkers. In the first study, which includes both English-French and English-Spanish translation pairs, participants are asked to determine whether there is a difference in meaning between the sentences in a translation pair. Here, participants who are given contrastive phrasal highlights do significantly better than those who receive no highlights, as measured against ground truth, and self-report that the highlights are useful. IAA is also higher, as measured by Cohen's kappa. In the second study, participants are asked to detect translation errors and rate their severity. Here, too, participants who are given contrastive phrasal highlights do better, and again self-report the usefulness of the highlights.

**Questions For The Authors:**

A. For the study in section 6, what exactly were the annotation possibilities per sentence? No error, minor error, major error? It seems from the text that you focused on major errors. It's thus unclear how exactly you define precision/recall/F1 for Figure 4. Please clarify this.

B. Please provide full details on what statistical tests you used in sections 5 and 6. It also seems that you didn't do any significance testing for section 4; are you able to provide this?

C. Figure 3 (and on line 433) you claim statistical significance but say p=0.1. Given the magnitude of the difference, I assume that this is a typo (and you meant, e.g., p=0.01), as you later  in Figure 4 say that a value of p>0.05 is *not* significant.

D. Lines 215–216: it's not clear what is meant by "unaligned words... can be included in either phrase". Does this mean that, if unaligned words are adjacent to an identified phrase, they are added to the phrase? What do "can" and "either" mean here?

E. Line 238: what is |S|? Is it |S| = max(|S_1|, |S_2|) where S = (S_1, S_2)?

F. Line 238: why is it meaningful to look for R(...) >= 0 if R is a ranker trained for pairwise ranking (as indicated on lines 190–191)? Should this perhaps be R(DEL[S;p]) - R(S) >= 0?

G. Line 238: the answer to the previous question might help resolve this, but as-is I'm not clear on why you increase the score in the 'otherwise' case.

H. Table 2: "Del." isn't explained. Judging from the text just below it this is the percentage of the words in the sentence that are highlighted. Is that correct?

I. Line 293: the random baseline isn't defined. Assuming I'm correct about "Del.", it looks like this just randomly highlights tokens with 50% probability. Is that correct?

J. Why no F1 scores in Table 2? (Assuming I'm correct about it, )"Del." is IMO less useful, and the brief discussion of it in the text just below Table 2 suffices.

K. Do you have data on time-to-completion for the tasks that your participants performed? That's another measure that could help support the hypothesis of the utility of contrastive phrasal highlights. For example, participants in the with-highlights condition might have been able to complete the task faster than those in the without-highlights condition, on top of the improvements in F1.

L. Line 547: you hypothesize that the WMT errors are more easily spotted. This seems intuitively plausible, but it seems that your data in Figure 4 contradict this: the without-highlights precision, recall, and F1 are very similar in the WMT and Negation cases. Is that assessment accurate?

M. Footnote 7: what do you mean by "corroborate" here?

N. Since you have 12 participants, please specify how you handle ties (if there are any) in computing the majority stats for Figure 3.

O. In section 5.3, it seems that you have multiple measures that reflect participants' "perceived understanding of the explanation", yet you give only one example. Were there others? How did the participants respond?

**Reasons To Accept:**

* The paper is well-written and easy to follow. Seriously! I have technical questions, sure, but most papers I review aren't as well-organized or as cohesive and coherent as this paper. For example, the hypotheses for the two application-grounded studies are clearly stated up front to help clarify what is being explored in the relevant section.
* The method presented is (mostly) easy to understand.
* The method presented is conceivably applicable to arbitrary models (the authors don't mention this, and I think it'd be worth mentioning as a strength in the paper!), and the use of pre-existing phrasal alignment methods means that the authors' method will benefit from advancements in phrasal aligners.
* The method presented improves on prior work and the human evaluations suggest that the contrastive phrasal highlights are legitimately useful.
* The authors tested their method with multiple language pairs.

**Reasons To Reject:**

Nothing I'm writing here is a reason to reject, really (though I have some questions in the "Questions For The Authors" section that may impact my sense of this). But there are some minor weaknesses:

* The method presented relies on external phrasal alignment models (this is also a strength as noted above!)
* Some details are unclear (see "Questions For The Authors")
* While the authors argue strongly (and, I think, correctly) that phrasal highlights are more useful than simpler token-based highlights, the human evaluations do not directly test this. It's conceivable that, for example, annotators would find the simpler token-based highlights just as useful as the contrastive phrasal ones. While I think this is unlikely, it undermines the overall argument somewhat. To be clear, I'm not especially bothered by this as the very fact that the authors included application-grounded evaluations is a useful contribution (and we can make some reasonable inferences based on the results from the proxy evaluations).
* There's no indication that the authors will release their code or the data presented to annotators for the studies in sections 5 and 6 (including annotations from the participants). I would highly encourage this, especially since it can help clear up details that might be unclear from the paper's main text (or even in appendices). Similarly, including full details of the ChatGPT prompts in section 5.1 (beyond just footnote 5) would help reproducibility.

**Reproducibility:**

3: Could reproduce the results with some difficulty. The settings of parameters are underspecified or subjectively determined; the training/evaluation data are not widely available.

**Reviewer Confidence:**

4: Quite sure. I tried to check the important points carefully. It's unlikely, though conceivable, that I missed something that should affect my ratings.

**Typos Grammar Style And Presentation Improvements:**

It may seem like I have a lot here, but I promise, I only provide this kind of feedback in my reviews when the writing is above a certain level, which you have very clearly surpassed! And yes, some of the comments here will be nitpicky...

* The colours do not come across well when printed in black & white, which is difficult to adjust for, but I do strongly recommend that you verify that your colour choices are appropriate for those with colour-deficient vision (colorbrewer2.org is a good place to start).
* There's no mention of translation as the primary task in the abstract; instead, more broad, general language is used, which gives the reader (or at least me) the impression that the work in this paper will be applicable to a wider range of tasks.
* Line 037: "show how they differ" comes across as circular here, or at least vague: "...to explain how [they] differ, it is more intuitive to show how they differ...". What exactly does "show" mean, and how is it different from what follows (i.e., "highlight all salient Greek and English tokens without specifying how they relate to each other" qualifies as "showing" in my mind... or at least as an attempt to show...). Obviously this becomes clear through the rest of the paper, but when I was reading this passage I had no clue how to interpret it. Ironically, what's missing here is a *contrasting* example. Figure 1 shows contrastive phrasal highlights, but the alternative is vague and unspecified. An accompanying example for comparison (or rather, contrast) would be greatly helpful here. This comes up again on lines 176–177: what does "highlighting tokens" mean here? What would it look like as an example against the aligned phrase examples that follow?
* Line 043: when I read "the two inputs" on this line, I thought, 'Which two inputs?' The easiest fix would be to remove the "the".
* Paragraph starting on line 183: this is an important paragraph, as it defines the type of model that your method will be working with, and thus scoping the applicability of your work. It might be worthwhile to make this importance clear by putting this paragraph in its own subsection.
* Figure 2: It would help to visually cue the reader that larger values are on top in step 2, e.g. via arrowheads on the lines
* Figure 2: the aspect described in the first paragraph of section 3.2 is not made clear in the figure
* Line 470: "mistranslation error" is very vague. Wouldn't all errors in a translation be mistranslation errors? Examples would help here, but at the least use a more specific error type.
* Line 518/footnote 10: again, an example that contrasts a minor error vs. a major one would help here.
* Abstract: 2x occurrences of "NLP" should be in small caps to be consistent with the small-caps typesetting of "NLP" and (most) other abbreviations in the main text.
* Lines 010–011: "phrase alignment guided" -> "phrase-alignment-guided" to match line 048
* Line 078: "model internal" -> "model-internal"
* Line 080: "the ability of these highlights" -> "these highlights' ability" (avoids a garden path)
* Line 105: "seeks" -> "sought"
* Line 148: "differs" -> "differ"
* Line 190: Use `\cdot` instead of period
* Line 191: First closing parenthesis looks like it's in text mode when it should be in math mode
* Figure 2 caption: "Explain by" -> "Explaining with", to match your paper title. (I actually think you can get rid of this so that the caption just starts with "Our approach...")
* Figure 2: English sentence in p_1: "served as" -> "served"
* Line 200: "sentence-pair" -> "sentence pair"
* Line 204: "prohibited" -> "prohibitive"
* Line 284: ";2" -> "; 2"
* Line 268: I assume this should be Table 2, not Table 4? (It'd be ideal to refer to the tables in numerical order; I think what is currently Table 2 could be moved over to page 4 without bumping its reference to page 5; that way the first table that you refer to is called Table 1, etc.)
* Line 291: I'd suggest removing the words from this line, as there's no need to repeat this information (it was just explained in the preceding two paragraphs)
* Line 297: "the token space" -> "token space"
* It's unclear whether you're using small caps for abbreviations only or you're also using it for system names (e.g., "Random", "Ours", etc.) and even those are inconsistent (line 303 has "Ours" typeset regularly, but it is in small caps in Tables 1 and 2; likewise, "Random" is in small caps on line 293 but is typeset regularly in Table 2). Personally I prefer to leave small caps for the abbreviations (including, e.g., BR in line 303 and Table 2), but whatever you do, consistency would be nice :)
* Line 303: use minus instead of dash (like in Table 2).
* Line 309: "help" -> "helps"
* Line 314: The sentence beginning "However, as highlighted..." is an unnecessary repetition of the same point made in the first paragraph of section 4—it doesn't really say anything new. I think you can delete this sentence and instead add a sentence or two of detail to the first paragraph of section 4, specifically about how exactly the proxy evaluations can be misleading (with examples as needed). If you do this, I think it'd make sense to move the last sentence of sentence 4 to be the last sentence of the first paragraph of sentence 4 and rewrite accordingly (e.g., "We shall turn to application-grounded evaluations in subsequent sections.").
* A quick googling suggests that the name of the model is "Divergent mBERT", not "divergentmBERT" as you use in several places in your paper.
* Line 355: "chatGPT" -> "ChatGPT"
* Line 355: "hard" -> "difficult" (if that is the intended meaning, anyway)
* Line 358: "All examples are validated by the authors ensuring that" -> "We validated all examples, ensuring that..."
* Line 363: "semantic" -> "semantically"
* Figure 3 caption: "with vs. without highlights" -> "with- vs. without-highlights"
* Figures 3 and 4: to be consistent, it'd be ideal to use the checkmark/x based labels for the conditions.
* Lines 368–369: "...across two language pairs, i.e., English-French and English-Spanish" -> "...across the two language pairs (English-French and English-Spanish)"
* Line 377: "conditions where" -> "conditions: one in which"
* Line 379: "in the other" -> "a second in which"
* Footnote 6: "conveys" -> "refers to"
* Footnote 6: "..."Changed" information..." -> "..."Changed" refers to information..."
* Footnote 6: Since Figures 5 and 7 are so far away, it might be preferable to refer to them as "Figures 5 and 7 in Appendix A.3"
* Line 433: "accurate, across" -> "accurate across"
* Line 434: "and language-pairs" -> "and both language pairs".
* Line 435: remove "the group that does not access them—" (the following X HIGHLIGHTS label is sufficiently clear!)
* Line 444: I appreciate the alliteration, but "Cohen's cappa" -> "Cohen's kappa". Also, I've always seen kappa values reported as decimals, so the values should be 0.51, 0.33, 0.66, and 0.52.
* Line 469: "(i.e., major vs. minor)" -> "(as major or minor)"
* Line 499: "we artificially" -> "artificially"
* Line 542: "the former" -> "negation errors" (there is no proper referent for "former" here)
* Footnote 10: "Figures 10 and 9" -> "Figures 9 and 10" (the cleveref package can help for stuff like this)
* Line 559: "user's" -> "users'"
* Lines 565 and 568: "example(s)" should be "Figure(s)". The subcaption package might help here. As I suggested for Footnote 6, since the figures are so far away, it'd help to indicate that the figures are in Appendix A.3.
* Lines 589–590: "in (human) translated" -> "in human-translated"
* Table 3: reinforce that the left results are for the task given in section 5 and the right results are for the task given in section 6
* Lines 610–611: "can be, in principle, port to any language pairs" -> "can, in principle, be applied to any language pair"
* Lines 618–619: "quality introducing" -> "quality, introducing"
* Line 633: "15 USD" -> "15 USD/h" or "15 USD per hour"
* I'm not sure how this might have happened, but your opening and closing double quote glyphs are mismatched everywhere except on lines 1055–1056.

---

> ### Author Rebuttal · Authors · 2023-08-28
>
> We thank the reviewer for the constructive feedback and the many useful suggestions and corrections. We will incorporate all the suggestions in our revised version.
>
> We start by addressing the primary concerns about reproducibility raised by the reviewer and then proceed to address their questions.
>
> &#8594; **Reproducibility concerns:**
> Upon publication, we will release any code, data artifacts, and interfaces produced in this paper. Additionally, we will expand on the prompts used for the ChatGPT experiments (detailed below).
>
> &#8594; **ChatGPT details:**
> Starting from a given English sentence, we asked ChatGPT to edit the sentence in a way that introduces a small meaning difference. Below, we include the different prompts we used:
> * “Can you edit a small phrase in sentence X to introduce a small meaning difference?”
> * “Can you delete/add a phrase in sentence X to introduce a small difference in meaning?”
> * “Can you edit a small phrase that makes sentence X more general/explicit?”
>
> We then reviewed the edits introduced by ChatGPT to make sure it indeed introduced small meaning differences. If we are not satisfied by a current edit we ask again and potentially specify what parts of the sentence we would like to be edited.
>
> &#8594; **Question A:** That’s correct. As shown in Figure 10, bilingual speakers are asked to first indicate whether the translation contains an adequacy error and then specify its severity, i.e., major or minor. The severity is, by default, set to N/A in case the translation contains no adequacy error. Precision/recall/F1 scores are calculated only against major errors, as those are the ones we have gold labels for. That said, if a translation contains a major error (per our gold labels) but is annotated as minor or as N/A by humans, it counts as a false negative.
>
> &#8594; **Question B:** We report statistical significance based on bootstrap resampling. We draw samples randomly from the collected annotations with replacement. The sample size is the same as the one of the collected annotations, while the number of resamples is set to 1000. For the first study (application-grounded evaluation I), which contains a total of 150 annotations per language, the significance level is p=0.1, and for the second study (application-grounded evaluation II), which contains a total of 250  annotations, the significance level is p=0.05.
>
> We did not include significance tests for our proxy evaluations as the Recall improvements we report over prior work are huge, i.e., 37-39 points, with similar Precision. However, we can confirm that all Recall improvements are significant, with a p-value of 0.001. Precision improvements are not statistically significant as expected, given the small difference from prior work.
>
> &#8594; **Question C:** Given the smaller sample size of the application-grounded evaluation I, the calculated significance yields a p-value of 0.1, as detailed in our response to B.
>
> &#8594; **Question D:**  The sentence means that “unaligned words can be included in either the source ($p_1$) or the target ($p_2$) phrases”. We will revise the wording to address the confusion.
>
> &#8594; **Question E:** $|S| = |S_1| + |S_2|$. Thanks for pointing this out; we will add this definition to the revised version.
>
> &#8594; **Question F:** Our search is implemented in two steps as defined in the equations of Lines 230 and 237. At first, we are extracting a set of candidate counterfactual instances based on $\tilde{P}$. This set consists of all phrases, the deletion of which increases the ranking difference $R(\mathrm{DEL}[S;p]) - R(S) \geq 0$, as you correctly outline in your question.
>
> Then, in the second step (Line 237), we search within $\tilde{P}$ for the counterfactual that not only increases the score $R(\mathrm{DEL}[S;p])$ but also does so in a way that deletes the shortest possible phrase. Therefore, Line 238, which you indicate in your question, does not really implement the search but instead defines the brevity reward that encourages the selection of short phrases when searching within $\tilde{P}$.
>
> This reward is operationalized as an exponential function ($e^{|x|}$)  used when ranking the candidate counterfactuals in $\tilde{P}$. To achieve this, we switch the sign of the exponent in the brevity function in order to consistently reward shorter phrases when multiplying with $R(\mathrm{DEL}[S;p])$ (which can have either positive or negative values).
>
> &#8594; **Question G:** See our response in F.
>
> &#8594; **Question H:** Yes, that’s correct. We will clarify this in our revisions.
>
> &#8594; **Question I:** Yes, that’s correct. We will clarify this in our revisions.
>
> &#8594; **Question J:** We will add F1 scores in our revised version. Our approach surpasses prior work based on F1, too. For reference, here are the scores for the English and French sides separated by dashes:
>
> * Random: 43 - 45
> * LIME: 41 - 40
> * SHAP: 41 - 39
> * Ours:  62 - 64
>
> &#8594; **Question K:** Yes, we have collected time-to-completion data; however, there are no significant differences between the completion times of the different user groups. One reason we have not seen differences in that matter might be that the users who see highlights spend some time familiarizing themselves with what highlights represent. However, it would be interesting to explore whether those results change once users become familiar with the highlights or receive more training as to what they are, which we leave for future work.
>
> &#8594; **Question L:** Thanks for pointing this out! You are correct that this hypothesis is not supported by Figure 4, as those scores ignore altogether adequacy errors annotated as minor. Our hypothesis comes from our observation that bilingual speakers tend to spot WMT errors as adequacy errors but rate their severity as minor. This led us to conclude that they are not being genuinely missed out. On the other hand, Negation errors are not only misperceived as minor but are entirely overlooked. To clarify this, we will include the full annotation results (including minor and major errors) in the revised version.
>
> &#8594; **Question M:** We will rephrase it to “emphasize”.
>
> &#8594; **Question N:** Each instance is annotated by 3 participants. That way, majority voting has no ties (see lines 401-405).
>
> &#8594; **Question O:** For  Section 5.3, we only collect feedback based on the statement, “Highlights were useful in helping me detect meaning differences in the compared texts.". For Section 5.4, we additionally ask: “I would like to use the highlights to help me detect critical errors".

---

### Official Review · Reviewer_uUif · 2023-08-11

**Soundness:** 3

**Excitement:**

4: Strong: This paper deepens the understanding of some phenomenon or lowers the barriers to an existing research direction.

**Paper Topic And Main Contributions:**

The paper “Explaining with Contrastive Phrasal Highlighting: A Case Study in Assisting Humans to Detect Translation Differences” focuses on the task of improving explainability in the task of detecting errors and mistranslations, particularly from a human-centered perspective. Centering around the importance of phrasal pairs rather than tokens, the authors introduce a new method to produce highlights to explain differences between pairs of translated text. The authors validate this method against existing methods through a series of designed evaluations, which suggest that the introduced method strengthens the ability for humans to detect fine-grained differences in translated texts.

**Reasons To Accept:**

The paper is well-grounded and includes a thorough overview of highlight and contrastive explanations, the evaluation of such explanations, and prior work performed on explaining differences in translated texts.

The contrastive phrasal highlighting also presents a novel contribution, also in the way that it prioritizes brevity. This process is well-explained and illustrated.

The design of the human-centered studies is transparent, and the results are evaluated for significance. These results include a qualitative analysis, examples of which are provided in the paper.

Regarding the studies, the main findings of each study is discussed, and supporting arguments from free-form feedback is provided.

**Reasons To Reject:**

The main weakness of the paper originates from the data used in the application-grounded evaluations, its small-scale, and the sample size of the study. The first study contains 60 translation pairs, while the second includes 50. Regarding sample size, the first study includes 24 (12 per language pair) participants, weakening the generalizability of the reported statistical significance.

In addition, the prompts possibly could bias user responses, such as that the highlights “are AI-generated and indicative of meaning differences”. Such terminology could induce bias based on the user’s opinion on AI-generated content.

Following the presentation of the results, the discussion does not extend beyond the presentation of the results, making it unclear what conclusions the authors draw from the interpretation of their results, in response to the stated hypotheses. A connection between the results of the separate studies would strengthen the analysis – for example, how can the results of second study, positing that bilingual crowd-workers are assisted by contrastive phrasal highlighted in detecting some errors, be connected to the results that the same highlighting assists in detecting semantic differences?

**Reproducibility:**

2: Would be hard pressed to reproduce the results. The contribution depends on data that are simply not available outside the author's institution or consortium; not enough details are provided.

**Reviewer Confidence:**

2: Willing to defend my evaluation, but it is fairly likely that I missed some details, didn't understand some central points, or can't be sure about the novelty of the work.

**Typos Grammar Style And Presentation Improvements:**

Line 102: seems like a word is missing here

Line 105: typo – “has seeks”

---

> ### Author Rebuttal · Authors · 2023-08-28
>
> We thank the reviewer for their time and feedback.
>
> &#8594; **Reproducibility concerns:**
> Upon publication, we will release any code, data artifacts, and interfaces produced in this paper.
>
> &#8594; **Sample size of application-grounded evaluations:**
> Given that prior work evaluates highlight-based explanations of (machine) translation errors based *solely on automatic evaluations*, we chose to initiate a human-centered discussion that evaluates the usefulness of such highlights directly with an in-depth exploration of multiple application-grounded evaluations. We chose a sample size of 50 instances for each of our three studies, following the experimental settings of prior NLP work on human-centered evaluation of explanations (for other tasks), which evaluate 20-100 instances (Schuff et al., EMNLP 2020) (González et al., Findings 2021)  (Xie et al., Findings 2022).
>
> We decided to split our budget, i.e., $500 in total, across different conditions to study how highlights can assist humans in detecting meaning differences in both human and machine translations across three language pairs. Besides the small scale of our studies, we report significant improvements and consistent patterns that indicate our contrastive phrasal highlights assist humans in detecting meaning differences across all three conditions studied.
>
> Finally, we agree with the reviewer that increasing the sample sizes of our studies would strengthen the significance of our results. We view our current findings as solid starting points for an in-depth exploration of the usefulness of highlights for human-centered applications that future studies can build upon and also extend with broad explorations.
>
> &#8594; **Possible terminology bias:**
> We agree with the reviewer that the wording around highlights presented to the users—in our case, “highlights are AI-generated and indicative of meaning differences”—could potentially bias the users’ responses. This is, however, true of any possible wording, and therefore, answering the question requires a study that explores alternative prompts and a per-user analysis.
>
> Although those are not questions we have asked in this paper, we can reasonably draw some inferences by looking at users’ agreement with correct AI and false AI predictions in our application-grounded evaluation I. As detailed in the paper, for this evaluation, we deliberately introduce semantically equivalent paraphrases that result in phrasal highlights that do not reflect meaning differences. Such cases allow us to draw some conclusions on whether bilingual works over-rely on highlights, i.e.,  do they blindly tag everything highlighted as having a meaning difference? Below are some additional statistics that suggest there are no (at least major) biases introduced by our terminology:
>
> * Agreement with correct AI: 90% (French) and 87% (Spanish)
> * Agreement with false AI:     33% (French) and 48% (Spanish)
>
> &#8594; **Response to stated hypotheses:**
> We accept both stated hypotheses as described in lines 439-440 and 534-536 for application-grounded evaluations I and II, respectively.
>
> &#8594; **Connection between different application-grounded evaluations:**
> Both of the application-grounded evaluations revolve around the annotation of meaning differences in translation pairs and show how the *same phrasal highlights* can assist humans in detecting both nuanced differences in human translation and critical errors in machine translations.
>
> Our first study establishes the correctness of our contrastive phrasal highlights more directly: bilingual speakers are asked to identify *any* small meaning differences in human translations. Our second study, building on the first one, investigates a broader scenario: bilingual speakers are asked to not just identify meaning differences in (machine) translations but also mark the ones that correspond to critical translation errors. The latter study reflects a broader application as our highlights are not tailored to a specific severity; therefore, both minor and major meaning differences can be highlighted.
>
> Our findings suggest that the usefulness of our phrasal contrastive highlights can be generalized to different use cases, as they help humans tailor their annotations to errors of different severity. This promising finding opens the room for broader explorations of how the *same highlights* can help humans detect different types of errors that might be specific to a language pair, a domain, or an application based on the use case.

---

### Meta-Review · Area_Chair_e8VQ · 2023-09-16

**Recommendation:** 4

**Metareview:**

The paper focus on how to better assist humans to identify translation differences, proposes a new approach, and performs detailed analyses and comparisons between different approaches.

All reviewers agree that this paper is sound and exciting. The paper is well-written and easy to follow.

But in the final version, the authors should include discussions on the scale of the data used in the application-grounded evaluations, as pointed out by reviewer  uUif.

---

### Decision · Program_Chairs · 2023-10-07

**Decision:**

Accept-Main

**Comment:**

The paper focus on how to better assist humans to identify translation differences, proposes a new approach, and performs detailed analyses and comparisons between different approaches.

All reviewers agree that this paper is sound and exciting. The paper is well-written and easy to follow.

But in the final version, the authors should include discussions on the scale of the data used in the application-grounded evaluations, as pointed out by reviewer  uUif.